# Enhancing food security sustainability through digital information extension services in rural Uganda: Maize postharvest evidence-based strategies

Jackline Estomihi Mayende Kiwelu[1]*, Patrick Ngulube[2]

**1** Library Department, Aga Khan University, Tanzania, **2** Department of Interdisciplinary Research and Graduate Studies, University of South Africa, Pretoria, South Africa

* jackline.kiwelu@aku.edu

## Abstract

### Background

Postharvest losses caused by poor drying and storage practices lead to maize waste, reduced food availability, unsafe food due to aflatoxin contamination, and income loss for farmers. This exacerbates food insecurity and threatens the livelihoods of rural communities. Leveraging digital solutions to provide quality maize postharvest handling information is critical to mitigating these challenges.

### Objective

This study examined maize postharvest handling extension information services provided by rural agricultural extension officers in selected districts of Uganda, focusing on how evidence-based practices supported by information sources, expert knowledge, and ICT infrastructure ensure the quality and relevance of the information delivered.

### Methods

A convergent parallel mixed-methods design was employed. Data were collected from 312 rural smallholder maize farmers, 22 extension officers, and four information officers. Qualitative data were analyzed thematically using ATLAS.ti version 24, quantitative data were analyzed in SPSS version 29 to generate descriptive statistics and conduct Pearson's chi-square tests.

### Findings

Agriculture extension officers applied digital evidence-based information practices to improve maize postharvest handling advisory services. The study reported that the most frequently used practices were asking (identifying farmers' information needs),

**Data availability statement:** The dataset is accessible through the Centre of Open Science through https://osf.io/a8qk9.

**Funding:** The author(s) received no specific funding for this work.

**Competing interests:** The authors have declared that no competing interests exist.

acquiring (gathering relevant information), appraising (evaluating information quality), and applying (guiding farmers' decisions). Aggregation (organizing information for accessibility) and assessment (evaluating the effectiveness of applied information) were less practiced. The study suggests a significant relationship between evidence application and maize postharvest handling practices (p = 0.002). No significant relationship between asking farmers' information needs and maize postharvest handling practices (p = 0.887). The study identified inadequate updated sources of information, insufficient skills in evidence-based practices, and information communication technologies.

## Conclusion

This study suggests that decision-makers update the evidence source and develop continuous professional development (CPD) training programs for rural agricultural extension officers to equip them with knowledge and skills in maize postharvest handling, evidence-based information practices, and new developments in ICTs.

## Introduction

Sustainable Development Goal Two (SDG 2) promotes food security and sustainable agriculture to end hunger by 2030 [1,2]. Despite global agricultural improvements, food insecurity persists in rural areas of developing countries, where smallholder farmers face numerous challenges. Postharvest losses of staple crops like maize remain a significant concern due to poor handling practices such as inadequate drying, improper storage, and ineffective pest control [3,4]. These losses reduce food availability, compromise food safety through aflatoxin contamination, and diminish farmers' income [3,5], exacerbating food insecurity and threatening livelihoods [3].

Although food availability is a key component of food security, the efficiency of postharvest handling plays a critical role in ensuring that food reaches the market or is available for consumption [1,2]. Unfortunately, studies in literature report that many farmers, particularly in low-resource settings, lack quality information on effective postharvest management [3–6]. Without proper knowledge, farmers cannot implement simple yet effective practices to reduce losses and preserve the nutritional value of their crops [7]. In the case of maize, a staple food crop for millions, this knowledge gap can have far-reaching consequences, reducing food security and hindering economic stability [3,7]. Further research is needed on the quality and sources of information provided to rural farmers and the use of digital evidence-based practices in agricultural extension. This study addresses this gap by examining extension officers' evidence-based practices.

Access to quality maize postharvest handling information is crucial to addressing food insecurity. [7]. Traditionally, agricultural extension involves sharing knowledge, skills, and technologies with farmers to improve productivity and sustainability [8]. It bridges the gap between research and practice, especially in developing regions,

by providing training, support, and access to innovations that enhance food security and resilience [8,9]. When provided through well-trained extension workers, this information can empower farmers to adopt better practices for handling, storing, and preserving their maize crops [8]. Extension workers, who act as intermediaries between research institutions and farmers, are key in disseminating critical information [10,11]. With proper training and resources, these workers can provide the necessary guidance on reducing postharvest losses, improving crop quality, and ensuring the sustainability of farming practices [8,9]. By ensuring that farmers receive up-to-date, relevant, and practical knowledge, extension workers can help bridge the gap between research and real-world application, ultimately contributing to greater food security and household income [12].

In Uganda, maize smallholder farmers face challenges in accessing evidence-based information (EBI) for effective postharvest handling, leading to uninformed practices and significant losses, including income loss [4,6,13,14]. The Evidence-Based Information Practice (EBIP) process can address this by integrating research into practical applications that empower users with reliable and relevant information [15]. However, despite efforts by Uganda's Ministry of Agriculture to provide information through the Department of Extension Services, access to EBI for postharvest handling remains inadequate, with issues such as mould, termites, and aflatoxin contamination continuing to undermine maize quality [6]. As a result, farmers continue to suffer from persistent postharvest losses and health risks. These challenges are closely linked to traditional practices such as harvesting maize before full maturity, drying cobs on bare ground, storing in poorly ventilated structures, and transporting in non-airtight bags for long distances. Such practices increase exposure to moisture, pests, and fungal infection. Providing farmers with effective digital agricultural extension information services is important to enhance food security across four dimensions Amin, Mehrez [16]. They improve availability by offering real-time guidance on post-harvest management and reducing losses [8]. Access is enhanced by connecting farmers to markets and providing market insights, boosting economic opportunities. In terms of utilization, digital resources educate farmers on nutrition, food safety, and proper food handling, ensuring food meets dietary needs [8]. Finally, stability is supported through knowledge on sustainable practices, risk management, and climate-smart farming, with digital platforms offering timely weather updates to help farmers adapt to climate change and ensure consistent food security [8].

Suppose agriculture extension officers have access to comprehensive and credible sources of information, such as digital agricultural information from academic journals, government publications, and research findings, Information Communication Technologies (ICTs) infrastructure to enhance the ability to access, share, and apply information. In that case, they are better equipped to apply evidence-based practices (EBP) in extension advisory services [16]. However, limited or outdated access and unreliable ICT infrastructure can hinder their ability to implement the most effective information practices, leading to inefficiencies and poor outcomes in agriculture practices [17]. When agricultural extension officers know the value of evidence-based practices, they are likelier to seek out and use EBP methods. Understanding the importance of EBP encourages extension officers to move away from unverified practices toward scientifically supported approaches [18] that are relevant to farmers needs, as it is encouraged by Nalweyiso, Mbabazi [19,20]. This awareness can be fostered through training in the form of workshops, webinars, and exposure to successful case studies where evidence-based methods have improved extension service outcomes [21].

In the digital era, the ability to access, use and share digital evidence effectively using ICT tools such as databases, mobile apps, artificial intelligence [22] and other online platforms has enhanced the implementation of EBP [23,24]. Extension officers with ICT skills can access, analyze, and share information more quickly and accurately [25–29]. Additionally, ICT can facilitate communication with farmers [30], enabling the timely dissemination of evidence-based advice [31]. Lack of ICT and EBP skills in the digital era may result in missed opportunities to leverage digital tools for improving practice and extending new knowledge to the farming community [30]. Agricultural extension officers who implement structured, evidence-based information practices ensure their advice to farmers is grounded in scientific and verifiable information [18]. These practices identifying farmers' information needs, gathering, appraising, aggregating, and applying information help improve farming outcomes, promote sustainability, and reduce risks.

This study was set to investigate how agricultural extension officers' access to sources of information, awareness of the need for EBP in agricultural extension information services, knowledge and skills, and practices influenced maize postharvest handling in Uganda. Furthermore, it explores the process and importance of access to quality postharvest handling information and the role well-trained extension workers play in providing this essential knowledge in Uganda. It examines how targeted training programs and extension services can reduce postharvest losses, improve nutrition, and enhance the overall sustainability of agriculture, all of which contribute to the achievement of SDG 2.

## Conceptual framework

**Evidence.** This study defines evidence as researched scientific information meticulously verified for quality, relevance, accuracy and credibility [32]. Scholars have argued that evidence-based information (EBI) originates from primary research, such as experiments, surveys and observations and secondary research, such as systematic reviews, and is grouped according to the level of filters or appraisal from high to low quality [32–34]). Evidence that is put into practice prompts actions, as in health science, where evidence is applied at the point of healthcare. Agriculture evidence in this study can be put into maize postharvest handling practices as actions.

**Evidence-based information practice (EBIP) framework.** EBIP is implemented in a step-by-step process that has evolved. Booth and Brice [35] described six EBIP stages: 1) define the problem; 2) undertake and appraise research; 3) prototype and test; 4) implement the solution; 5) evaluate the outcomes; and 6) Engage in storytelling. Koufogiannakis [36] proposed six concepts, which are popularly referred to as the '6As': ask, acquire, appraise, aggregate, apply and assess. Thorpe and Howlett [37] developed four phases of EBIP: 1) interpret the organisational context and strategic priorities; 2) apply the institution's strategy; 3) measure the outcomes; and 4) communicate the impact. Furthermore, Thorpe [38] said EBIP is to articulate, assemble, assess, agree/apply and adapt. EBP originated in health sciences and is more implemented in health information sciences and minimally in agriculture information science. The main goal of EBIP is to provide a quality information service that is effective and efficient and satisfies both the farming communities and the agriculture extension service providers, as was explained by Hasanpoor, Bahadori [39], who evaluated the challenges and facilitators of EBP in health consultation services.

This study tested the 6As [40] in the context of Uganda and agriculture extension information services, specifically maize postharvest handling information, and found them applicable and relevant in evaluating information for relevancy and quality service and practices. The EBIP components (ask, acquire, appraise, aggregate, apply, and assess) flow from one step to another. It works with the facilitators like sources of evidence, experts' knowledge, ICTs, and Social values. The feedback gained during the 'assess' step could either be areas of improvement or another need discovered, as well as success stories during the EBIP (Fig 1).

The EBIP process is continuous, as shown in Fig 1.

1. Ask: This involves looking for a specific need of the problem for the relevant evidence. In this study, the evidence needed was identified from maize smallholder farmers in relation to the goal, which was improved postharvest handling. Defining specific needs aids in eliminating information needs that are not relevant to specific postharvest handling needs and create focus [36]. A well-defined question provides proper guidance for the remaining EBIP processes and yields a good outcome.

2. Acquire: This is searching for the needed evidence from various sources identified as relevant, such as a database, a repository, or an individual's website with the needed evidence. The evidence can be facts, practice or tangible items. In this study the extension services should involve farmers to know the sources of the information where they themselves can use to get relevant information.

3. Appraise: The acquired evidence is filtered to get the most relevant and valuable evidence to answer the identified evidence under stage one (ask).

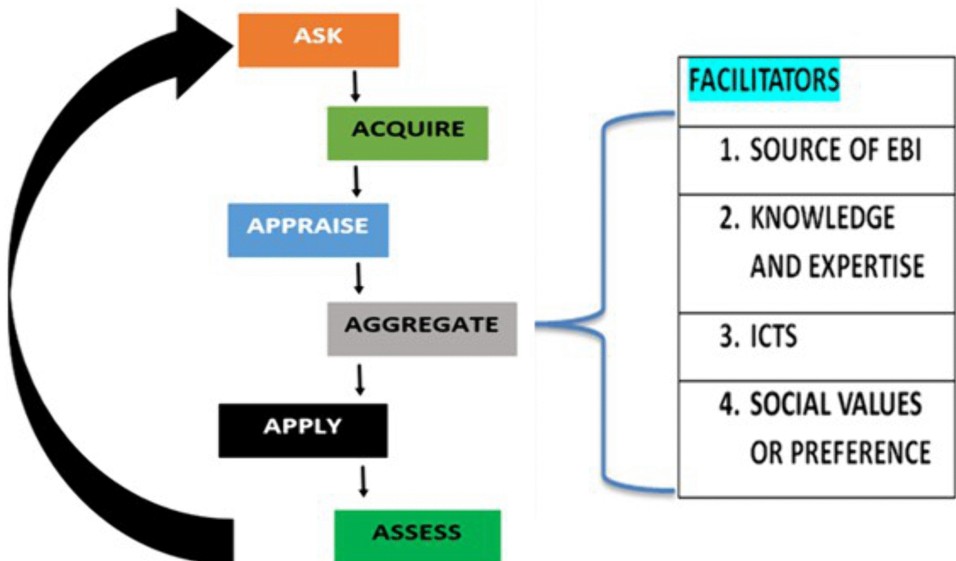

**Fig 1. Evidence-based information practices process (Source: Researchers, 2025).**

4. Aggregate: This is the process of gathering appraised evidence from various sources, translating it, and packaging it in an easy-to-use package. It is done according to the social context of rural smallholder farmers.

5. Apply: Use the appraised evidence to solve the problem identified in stage one (Ask). This can involve farmers receiving training, generating tangible items like maize storage or drying equipment, or using the facts for specific postharvest handling decision-making.

6. Assess: Evaluating and monitoring the status of the specific maize postharvest handling problem to which the evidence was applied to measure effectiveness and identify areas for improvement. These areas for improvement become new postharvest handling evidence needs, which start the EBIP process again.

To leverage EBIP for effective agriculture extension information services, facilitators are the source of evidence, expertise, social value and ICTs. The sources of maize postharvest handling evidence can be databases, repositories, libraries, manuals and the like. The knowledge and expertise of agriculture extension officers, researchers, and information officers are required and considered in the EBIP [39]. Farmers' social values, norms, ethics and socioeconomic status should be considered when implementing EBIP [39]. EBP involves a doctor or health practitioner, research evidence, and patient preference in medicine or health science. Similarly, in EBIP implementation in agriculture extension services, the agricultural extension workers must work with other stakeholders like information officers, farmers, and researchers or technology experts. With the evolution of ICTs, the process of EBP has been enhanced. ICTs are a combination of network infrastructure, hardware and software that facilitates information generation, storage and transfer [41]. Through ICTs, the world has been connected. It has become a 'global village', which means information can easily be generated, stored, and communicated from one part of the world to another via the Internet.

## Literature review

Society that can exploit ICTs capabilities [30,42] can access and use EBI more quickly today than it could 20 or 30 years ago when ICTs, including the Internet, were not available. Today, mobile phone technology [31] and television have taken

root in rural areas in developing countries, including Uganda. The [43] reported that 74% of the Uganda population owned a mobile phone, and 86% of phone owners used it to access Internet and social media services. [5] studied ICTs and agriculture information access and ascertained that ICTs could influence access to information for managing agriculture production.

Artificial intelligence (AI) is an emerging computer-based system designed to perform tasks that typically require human intelligence and reasoning, such as problem-solving and decision-making [44]. AI is significantly reshaping information dissemination not only in urban but also in rural areas [26,27,45–52].

Innovations in AI, combined with technologies like blockchain, chatbot and AI-generated content (AIGC) [26,52,53], enhance EBI communication, improve access to EBI resources, and foster socio-economic development in rural communities worldwide. One notable advancement is the integration of blockchain technology with AI, particularly in rural live broadcasting, which has significantly boosted information dissemination [27,46,48]. AI-generated content is transforming how multimedia information is shared by exploiting algorithms for specific information needs, enhancing user engagement, and providing timelier and relevant farming and other information [25,52,54]. If harnessed properly, these technologies will reduce the barrier to accessing agriculture extension advisory information services for rural farming communities. AI and related ICT technologies hold immense potential to revolutionise information dissemination in rural areas, enhance access to EBI resources, empower communities, and support postharvest handling and overall agricultural development. Chatbots empowered by artificial intelligence and natural language capabilities can search, appraise, and aggregate EBI information for rural farmers. These technologies can understand farmers' information needs, translate the needs and provide localised feedback.

Furthermore, AI research on rural development in the farming sector, particularly in machine learning and digital agriculture, has increased [52]. International collaborations between China, the U.S., and India are forming to leverage AI for rural empowerment, highlighting the global interest in using AI to address rural challenges [52]. AI-enabled systems like intelligent agricultural information systems (Agr-IS) are crucial in improving decision-making and providing EBP. These systems, which integrate ICT and e-government frameworks, help bridge the gap between farmers and governmental agencies, providing timely and accurate information that empowers farmers to manage farming practices [55,56]. For instance, in rural India, AI-driven devices are helping individuals with limited literacy skills access important information and other services [28]. In Tanzania, Platforms like MkulimaGPT also offer real-time farming advice via chatbots, helping overcome language barriers and ensuring farmers have timely access to critical information [53].

AI applications are increasingly designed with inclusivity, ensuring that marginalized groups in rural areas have equitable access to information and resources [29]. Despite these advancements, several challenges persist. Infrastructural limitations, socio-cultural barriers, and policy constraints continue to hinder the full potential of AI and ICT technologies in rural areas [30]. Digital illiteracy and inadequate infrastructure must be addressed to ensure that rural communities and agricultural extension officers can fully benefit from these innovations.

However, overcoming challenges related to infrastructure, socio-economic barriers, and ethical considerations is crucial to unlock their full potential. Addressing these issues will ensure that AI and ICT can be leveraged equitably, within the rural African norms, culture, ethics and sustainably, benefiting all rural populations and contributing to long-term rural development [49,50,57,58]. Abate, Bernard [59] and Mwantimwa [31] indicated that ICTs were necessary technologies to enhance and use to promote adequate access to agricultural knowledge that can be appraised, aggregated, and applied for effective agricultural information services. ICTs, including artificial intelligence capabilities, can enhance farmers' access to and use of agriculture EBI.

Furthermore, a pilot study on using information kiosks to improve rice farming in Thailand found that the kiosks effectively support farmers' access to and use of online and offline rice farming knowledge and skills [41]. That study suggested that the same method could help to support other crop farming in rural communities in Uganda. [60] there was close relationship between access to evidence and effective agriculture practices like maize postharvest handling.

Development partners such as the FAO, UNESCO and the World Health Organization have supported access to scientific and technological information for developing countries through Research4life resources in five collections: Health InterNetwork Access to Research Initiative (HINARI), Access to Global Online Research on Agriculture (AGORA), Online Access to Research in the Environment(OARE), Research for Development and Innovation and Research for Global Justice [61]. This support has improved access to and use of scientific research, especially in academic and research institutions in these parts of the [62]. AGORA archives scientifically researched agricultural information published by reputable researchers for agriculture practices like maize postharvest handling. Community, academic and research libraries have free or low-cost access to disseminate these resources to their users. However, whether farmers access and use AGORA collections as a source of EBI is unclear. Furthermore, whether agriculture extension workers use AGORA to equip themselves with scientific and technological information to transfer to farmers for improved maize postharvest handling was unclear in the literature. However, the use of AGORA EBI by agriculture extension officers to implement EBIP for effective agriculture extension information services remained unknown in the literature. The present study contributes knowledge in this area.

**Specifically, the study was guided by four objectives**

1) Examine how awareness and access to reliable sources of information influenced the implementation of evidence-based practices in agricultural information extension services.

2) Describe how information and communication technologies (ICTs) influenced implementing evidence-based practice in agricultural extension services.

3) Examine the contribution of evidence-based information practices in agricultural extension services.

4) Identify challenges and recommendations for improved agricultural information extension advisory services.

**Hypothesis tested from the quantitative data set**

This study formulated hypotheses for the six 6As, with the null hypothesis stating that no significant relationship existed between each EBIP and the maize postharvest handling. The researchers employed Pearson's chi-square product-moment correlation to test these hypotheses, which allowed us to examine whether statistically significant associations were present.

## Materials and methods

### Study design and setting

This study used a mixed methods approach combining quantitative and qualitative techniques. A concurrent, convergent design was adopted. The participants included 22 agricultural extension officers, four information officers and 312 rural maize smallholder farmers from Adjumani, Iganga, Kabarole, Mubende, and Entebbe Municipality in Uganda.

### Data collection

Quantitative data were gathered using a structured questionnaire. The instrument consisted of both closed and open-ended questions organized into five sections: (a) Background information, (b) status of maize postharvest handling, (c) access and use of EBI, (d) recommendations.

The questionnaire was administered only to rural maize smallholder farmers. The complete questionnaire is included as Appendix 1. Qualitative data were collected through personal interviews, focus group discussions, open-ended responses, and document analysis of government policies. The study was carried out from November 2023 to May 2024. The use

of multiple groups of participants (i.e., smallholder farmers, agriculture extension officers, and information officers) and multiple concepts (maize postharvest handling, EBIP, smallholder farming, extension agriculture information services) was for corroboration and complementarity [63].

The methodological process followed is summarised in Fig 2.

Fig 2 shows a cross-sectional survey using a questionnaire on random stratified sample respondents from rural smallholder farmers and an in-depth interview using a semi-structured guide on purposively selected agriculture extension officers and information officers. Government documents were also analysed, and two focus group discussions were held. Quantitative and qualitative data were collected simultaneously and analysed separately; integration happened during interpretation, discussions, conclusions and recommendations. The two data types are collected parallel for validation and comparison before conclusions are drawn [65]. The research data was collected from November 2023 to May 2024. The researchers kept the participants' information and contacts and verified the data with the participants during and after the data collection.

## Data analysis

The qualitative and quantitative data were analysed separately and then merged during the interpretation phase, as advised by the mixed methods research experts [64,66]. Quantitative data analysis involves extracting meaning, relationships, and trends from numerical data collected in surveys. Statistical software like SPSS (version 29) was used for analysis. Descriptive statistics, including frequencies and percentages were done. Pearson's chi-square product-moment correlation was used for correlational analysis to examine relationships between categorical variables. A five-level scale (1–3.90) was employed for responses. To assess statistical significance, chi-square tests are used at a 5% significance level, with cross-tabulations, likelihood ratio, and linear-by-linear association analyses offering further insights into variable relationships. Degrees of freedom (df) indicate the number of independent data points used in these tests. Asymptotic

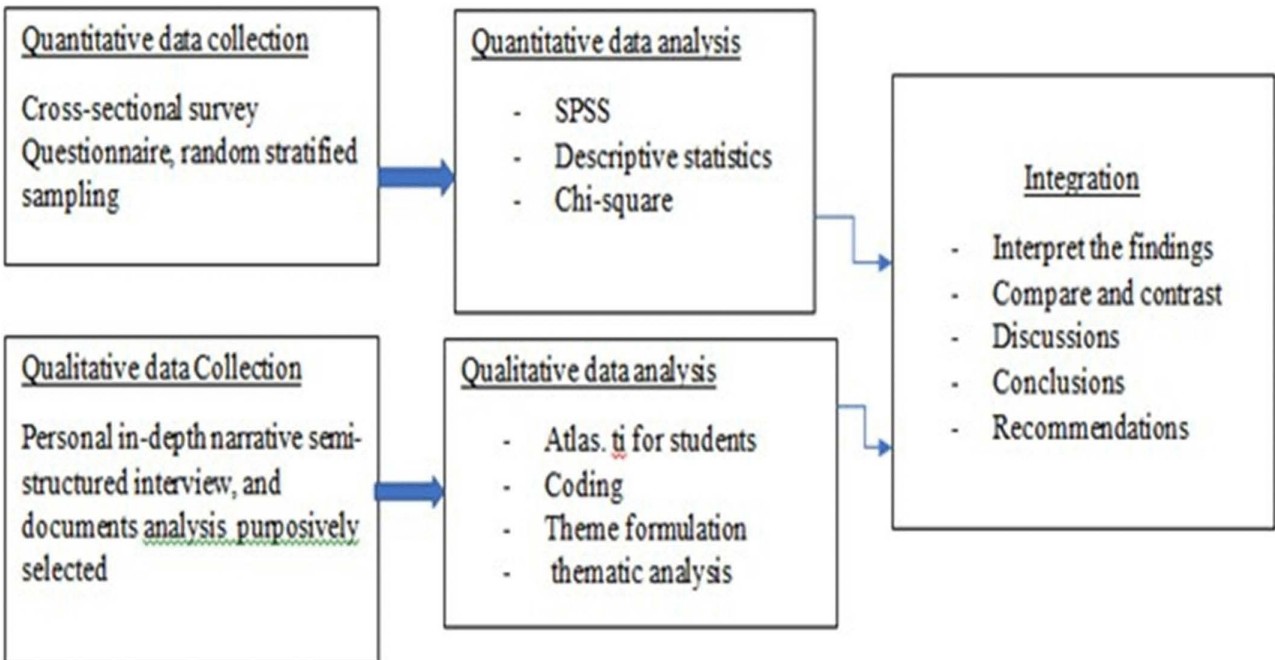

**Fig 2. Convergent parallel mixed methods design.** Source: [64].

significance (p-value) shows the likelihood of observing the test statistic under the null hypothesis. A p-value above 0.05 means no significant relationship is found, while a p-value below 0.05 indicates a significant relationship between the variables.

Qualitative data analysis for this study, the six-phase thematic analysis framework by Braun and Clarke [67] includes knowing the data, coding, forming themes, reviewing themes, deciding on key themes, and reporting results. Qualitative data were analyzed using ATLAS.ti 24, a computer-assisted qualitative data analysis software that supports systematic coding, organization, and retrieval of textual and visual data. ATLAS.ti 24 was used to code, categorise themes, generate visual figures, frequency tables, word clouds, and networks, and observe meaningful comparisons, associations, and correlations. The researchers uploaded the data to ATLAS.ti 24 in the form of documents. The uploaded documents were grouped into five groups based on how the findings were realised (personal interviews, focus group discussions, quantitative open-ended questions, observations and document analysis). The researchers read and re-read the documents one after another to become familiar with the data. The most relevant segments of the data were identified as the researchers read and re-read the documents. Next, the first codes were generated by marking the relevant segments in the documents as quotations and assigning codes using open and quick codes. The codes were then grouped by their respective families or relevant overarching codes to help the researcher organise the findings and understand the relationships between concepts from the perspective of the research questions and objectives. Thereafter, themes were reviewed and re-grouped, and some additions were made. They were then confirmed as suitable themes using common patterns, topics and connections as advised by Creswell and Poth [68], whereby similar codes formed code groups and refined code groups formed themes.

### Integration of the findings

In this study, convergent data integration was reached when one dataset confirmed the other (corroboration), one dataset explained the other (elaboration), one dataset introduced the other (initiation), both datasets complemented each other (complimentary), and both datasets contrasted each other. Integrating data promotes methodological transparency [69] and provides a comprehensive understanding that can yield deeper explanations and new insights than separate datasets [70]. The study objectively drew inferences using descriptive statistics and content analysis procedures based on comparing, confirming, explaining, and contracting research questions. A narrative weaving approach was used to explain the interpretation and reporting of the findings. This weaving approach presented the qualitative and quantitative findings on a theme-by-theme or concept-by-concept basis to show collaborations and relationships [71].

### Ethical clearance

The study was subjected to human clearance ethical reviews and received low-risk ethical clearance on 24/04/2022 from the College of Human Sciences Research Ethics Review Committee at the University of South Africa (UNISA), under registration number REC 240616 CREC_CHS_2022. All participants provided written consent before sharing their views and opinions. The researchers confirm that the study adheres to the principles of the Helsinki Declaration concerning research.

This study acknowledges using Sci Space to discover relevant journal articles and ChatGPT Open AI OpenAI 2015–2025 to rephrase.

### Findings

**Response rate.** The quantitative survey response rate was 78% or 312 of the 400 respondents, while the qualitative data set received 26 participants, 54.15% or 26 of 38 participants, as seen in Table 1.

In Table 1, Adjumani district (Northern region) had a response of 100%, followed by Iganga (Eastern Region) with 85.7%, Mubende (Central Region) with 77.7% and Kabarole (Western Region) with the lowest response rate at 54.3%

**Table 1. Response rate for Quantitative and qualitative data sets.**

| No. | District/Station | Sample | % of Sample | Response (n) | Response proportion % |
|---|---|---|---|---|---|
| 1 | Iganga | 133 | 33.20 | 114 | 85.7 |
| 2 | Kabarole | 105 | 25.50 | 57 | 54.3 |
| 3 | Mubende | 94 | 23.50 | 73 | 100.0 |
| 4 | Adjumani | 68 | 17.00 | 68 | 100.0 |
| | Total | 400 | 100 | 312 | 78.00 |
| **QUALITATIVE STUDY (In-depth Interview)** | | | | | |
| 1. | Entebbe Head Office | 6 | 12.50 | 2 | 33.3 |
| 2. | Adjumani | 11 | 22.91 | 7 | 63.6 |
| 3. | Iganga | 10 | 20.83 | 4 | 40.0 |
| 4. | Kabarole | 11 | 22.91 | 9 | 81.8 |
| 5. | Mubende | 10 | 20.83 | 4 | 40 |
| | Total | 48 | 100.0 | 26 | 54.15 |

for the survey. The low response rate in Kabarole was attributed to the poor road terrain that made data collection and callbacks difficult compared with other districts. The responses represented the rural smallholder farmers derived from Uganda's four regions.

Furthermore, the response rate for the personal interviews was higher in Kabarole (9, 81.8%) compared with the remaining districts (Adjumani, Seven, Iganga four, Kabarole four, and Entebbe Head office, two). The two focus group discussions attracted 15 participants in total. The first physical focus group discussion (FGD) had nine participants, while the second online FGD had six participants.

Documents from the Government of Uganda analysed to supplement the primary data include: i) Uganda National Household Survey 2019/2020 report (UNHS 2019); ii) National Agricultural Extension Strategy 2016/17–2020/21 [72]; iii) Extension Guidelines and Standards 2016 (EGS 2016); iv) The Constitution of the Republic of Uganda 1995 [73]; v) Uganda Information Access Act 2005 (UAA 2005; vi) the BMAU briefing paper 25/19, 2019 [74]; and vii) The Maize Training Manual for Extension Workers in Uganda (MTEWU) [75]. The information generated from these documents is presented with the other findings.

### Sources of agricultural maize postharvest handling practice information

The sources of agricultural evidence-based information are important for leveraging evidence-based practices in agricultural extension services. This study examined the sources of evidence for agricultural maize postharvest handling practices. The questionnaire respondents were asked to state where they accessed maize postharvest handling information, allowing multiple responses (Table 2).

**SACCO**: Savings and Credit Cooperative Organization.

Table 2 presents the quantitative findings, which indicate that 'friends' were the most dominant source of information for smallholder farmers (265, 84.9%), followed by village meetings (254, 81.4%) and the Agricultural Extension Office (248, 79.5%). Community libraries were the least commonly used information sources (34, 10.9%), and 1.3% of respondents did not know any information source.

The personal interviews and document analysis revealed that agriculture extension information services were accessed through farmers' groups, training, media and farm demonstrations. The sources of maize postharvest handling evidence for extension officers were the Internet, textbooks, AGORA, the Ministry of Agriculture, Animal, Industry and Fisheries (MAAIF), workshops, partners and classroom knowledge. Few participants acknowledged having used AGORA. When probed further, those who had used AGORA clarified that they used this source when studying at the university, which was

**Table 2. Sources of maize postharvest information by rural farmers (N = 312).**

| Sources of information | Number | % |
|---|---|---|
| Village meetings/SACCO and church meetings | 254 | 81.4 |
| Agricultural Extension Office | 248 | 79.5 |
| Non-governmental organisations | 215 | 68.9 |
| Personal experience learned offline or online means | 109 | 34.9 |
| District/village/sub-county information officers | 132 | 42.3 |
| Community libraries | 34 | 10.9 |
| Friends | 265 | 84.9 |
| Media (radio/TV/newspapers) | 187 | 59.9 |
| I do not know any source | 4 | 1.3 |

challenging in rural areas with limited Internet connectivity. Moreover, these participants felt they needed to widen and update their sources of evidence.

The participants interviewed further explained that extension officers did not work with the community libraries, librarians or district information officers. They cited not being aware of their ability to disseminate technical maize postharvest handling evidence to farmers. Some participants recommended that community libraries be located near rural smallholder farmers and actively introduce themselves to the farmers. In contrast, participants from community libraries explained that they were disseminating information to rural smallholder farmers using Internet services via the community library reference services.

One participant said, '*Every time farmers come to the library and the Internet is available, I help them, but if not, I cannot help*' (P21-IGA). Participants cited the lack of postharvest handling evidence in the libraries and skills on how to help farmers (P23-IGA). They suggested that extension officers work with community libraries by equipping them with packaged evidence to disseminate to the farmers who visit their libraries. However, the National Agricultural Extension Services Strategy 2016/17–2020/21 did not mention community libraries as partners in disseminating information to farmers.

The qualitative arm of this study revealed that extension workers used the Internet, MAAIF Manual and books as sources of EBI related to maize postharvest handling. They disseminated this information to rural smallholder farmers through training, farm demonstrations, radio talk shows, farm visits and various events.

The qualitative findings from focus group discussions, document analysis and personal interviews the study suggest that there were five kinds of evidence disseminated to rural farmers namely i) Explicit information is found in databases, the Internet, research bodies' websites and other collections. ii) Implicit or tacit knowledge from agriculture experts. iii) Farmers' experience through practices that had worked for them over time. iv) Technological materials. v) Equipment found in factories or industries.

## Knowledge and skills in information and communication technologies (ICT)

The ability to effectively use ICT tools such as databases, mobile apps, and online platforms dramatically influences the implementation of EBP. Extension officers with strong ICT skills can access, analyze, and share information more quickly and accurately. Additionally, ICT can facilitate communication with farmers, enabling timely dissemination of evidence-based advice. Lack of ICT proficiency may result in missed opportunities to leverage digital tools to improve practice and extend new knowledge to the farming community.

The findings from both the qualitative and quantitative datasets suggested that ICTs had not reached all parts of Uganda, and some rural smallholder farmers could not access EBI online, even via WhatsApp. The document analysis showed that the Uganda National Extension Strategy 2016–20/21 recognized the capabilities of digital information and

ICTs to improve extension services. However, this study did not find proof that the agriculture extension officers implemented this strategy.

## EBIP and agriculture extension maize postharvest handling information

First, the relationship between respondents' access to postharvest handling information and demographic characteristics (N = 312) was examined. The chi-square test results, and p-values are shown in Table 3. A p-value above 0.05 means no significant relationship is found, while a p-value below 0.05 indicates a significant relationship between the variables.

**SACCO**: Savings and Credit Cooperative Organization.

The results in Table 3 show that the findings did not provide sufficient evidence to show significant relationships between gender and respondents' access to maize postharvest information as an aspect of EBP. However, the study provides evidence that SACCO (P-value of 0.022), Literacy level (P-value of 0.003), Age bracket (P-value of 0.003), and access to maize postharvest information have significant relationships.

**EBIP Expertise.** To ensure the information accessed is credible, of quality, and relevant, it undergoes an EBIP process, which includes the 6As: ask, acquire, appraise, aggregate, apply, and assess. Before something is used, people must know its existence and how it is used. When agricultural extension officers know the value of evidence-based practices, they are likelier to seek out and use EBP methods. Understanding the importance of EBP encourages extension officers to move away from unverified practices toward scientifically supported approaches. The study investigated the EBP awareness of rural farmers and rural extension officers. Therefore, respondents were asked to mention the EBIPs they knew were implemented in their maize postharvest handling practices. The survey results attracted multiple responses, as presented in Table 4.

The quantitative results (Table 4) showed the responses to question 19, which asked the rural small holder farmers, "What Evidence-based Information Practices are you aware of?". Responses were either Yes or No, as seen in Table 4, representing the total number of farmers who confirmed that extension officers performed the EBIP. 'Asking' was part of the EBIP process that was the most implemented (252, 80.8%), followed by appraising the evidence (174, 55.8%) and acquiring the information (147, 47.1%). Only 28 (9.0%) respondents said they were aware that the assessment of the

**Table 3. Access to EBI and respondents' demographics.**

| Gender | Response | % | Chi-square | P-value |
|---|---|---|---|---|
| Female | 134 | 42.9 | 2.243 | 0.291 |
| Male | 178 | 57.1 | 2.243 | 0.291 |
| Total | 312 | 100 | | |
| **Literacy levels** | | | | |
| I can read and write | 234 | 75.0 | 16.686 | 0.003 |
| I cannot read or write | 78 | 25.0 | 16.686 | 0.003 |
| Total | 312 | 100 | | |
| **Age bracket, years** | | | | |
| 18–40 | 240 | 76.9 | 23.691 | 0.022 |
| 41–60 | 67 | 21.5 | 23.691 | 0.022 |
| ≥61 | 5 | 1.6 | 23.691 | 0.022 |
| Total | 312 | 100 | | |
| **SACCO** | | | | |
| Yes | 154 | 49.4 | 15.713 | 0.003 |
| No | 158 | 50.6 | 15.713 | 0.003 |
| Total | 312 | 100 | | |

**Table 4. EBIP implementation and maize postharvest handling decision and practice (N = 312).**

| Evidence-based information practice (6As) | Yes | No | Chi-square | P-value |
|---|---|---|---|---|
| Asked about the farmer's information on maize postharvest handling | 252 (80.8%) | 60 (19.2%) | 0.020 | 0.887 |
| Acquired maize postharvest information | 147 (47.1%) | 165(52.9%) | 0.784 | 0.376 |
| Appraised maize postharvest handling information for credibility and quality | 174(55.8%) | 138(44.2%) | 2.712 | 0.100 |
| Aggregated maize postharvest handling information for easy access | 47 (15.1%) | 265 (84.9%) | 0.231 | 0.361 |
| Applied maize postharvest handling information for better practices | 90 (28.8%) | 222 (71.2%) | 9.188 | 0.002 |
| Assessed maize post-harvest handling information application | 28 (9.0%) | 284 (91.0% | 0.644 | 0.422 |

applied information was done. Findings from the personal interviews revealed that the most well-known and implemented EBIP components were asking, acquiring, and applying. The qualitative data source used A Sankey diagram to show the discussions on the relationship between EBIP and maize postharvest handling (Fig 3).

The qualitative arm of this study widely highlighted EBIP for maize postharvest handling, including its practices, the evidence disseminated, the way EBI was accessed and used, the information and communication technology, the digital content, various maize postharvest handling evidence needs, and the skills (Fig 3). This Sankey diagram indicates the source of the findings drawn from the focus group discussions, personal interviews, and document analysis. These findings showed a mixture of sources for verification and validation purposes.

Question 23 asked the respondents whether EBIP influenced their maize postharvest handling decisions and practices. The respondents who said 'YES' 237 (76.0%) meant that the EBIP influenced their decisions and practices, while those who said 'NO' 75 (24.0%) suggested that the EBIP did not influence them. The researchers cross-tabulated the findings of question 19 and question 23 to measure the significance level using Pearson's Chi-square. The findings are presented in Table 4.

The findings in Table 4 indicate no significant relationship between asking farmers' information needs and postharvest handling decisions and practices (p = 0.887). Therefore, this study failed to reject the null hypothesis and concluded that no significant relationship exists between asking farmers' needs and access to relevant evidence for maize postharvest handling. The null hypothesis was not rejected as there was no significant relationship between appraising evidence and postharvest handling decisions and practices (p = 0.192). The results showed that the null hypothesis was not rejected because there was no significant relationship between aggregating evidence and postharvest handling decisions and practices (p = 0.361). Finally, this study failed to reject the null hypothesis as there was no significant relationship between assessing evidence and postharvest handling decisions and practices (p = 0.422). However, the results indicated that the null hypothesis was rejected because, as illustrated in Table 4, a significant relationship existed between the application of evidence and postharvest handling decisions and practices (p = 0.002).

Furthermore, the study examined the difficulties that rural agriculture extension officers were experiencing. The quantitative and qualitative respondents mentioned that the significant difficulties encountered were inadequate resources, the high farmer-to-extension worker ratio, and rural smallholder farmers' poor mindset and unwillingness to adopt new maize postharvest handling evidence. One participant noted that the work of extension officers was meant to be in the field, disseminating information rather than being office-based; however, extension workers were mostly (80%) in offices and only spent 20% of their time in the field. Therefore, the impact of EBI was minimal. Suggestions were to increase the number of extension officers and improve road networks for better and easier movement to disseminate maize postharvest handling EBI to the rural population. Questionnaire respondents further recommended improved access to EBI among rural smallholder farmers, extension workers and information officers. The importance of retooling and updating extension and information officers with ICT knowledge and skills for packaging and disseminating information, better facilitation and more support for smallholder farmers was emphasised.

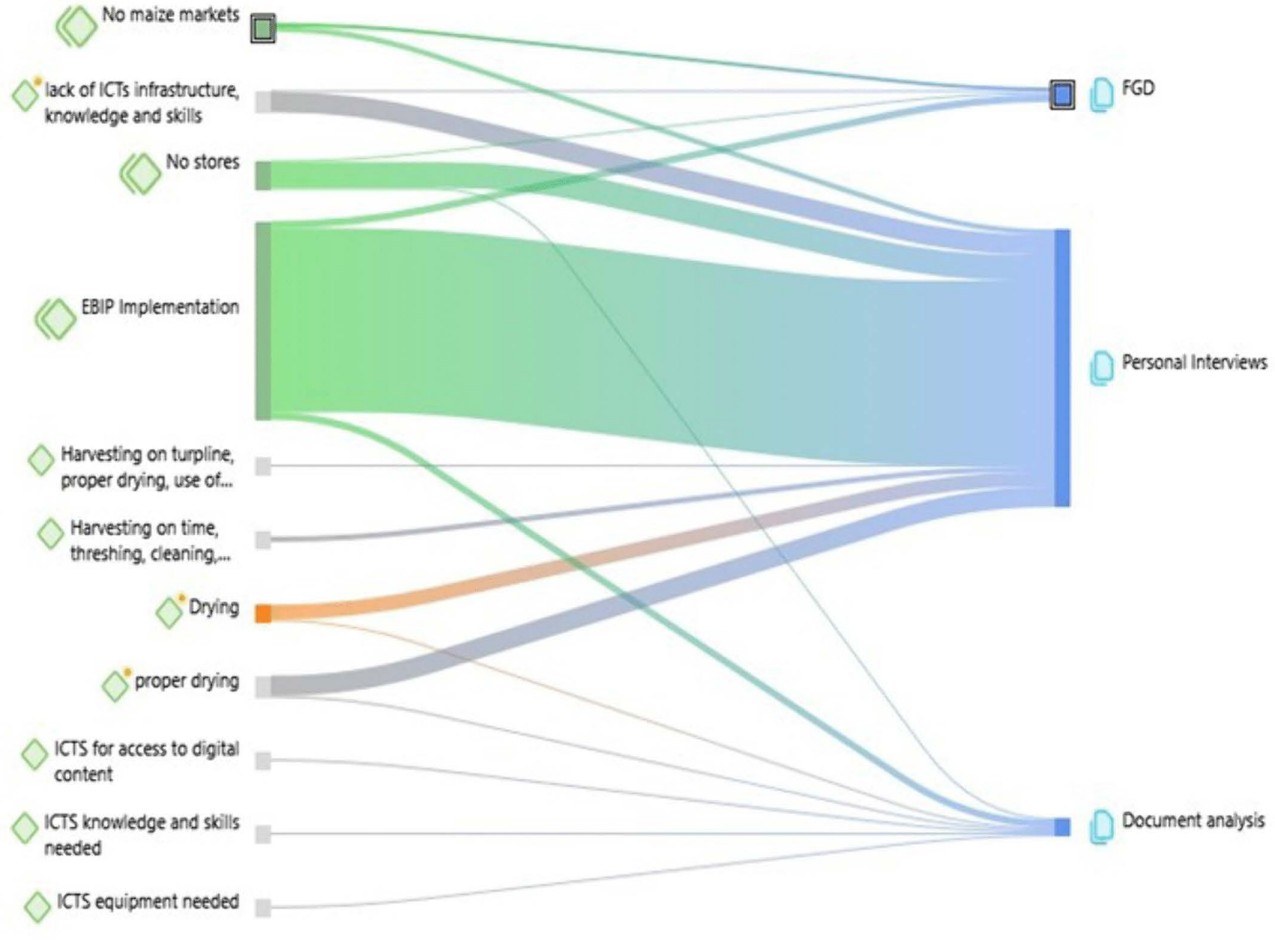

**Fig 3. Relationship between evidence-based information practice and postharvest handling decisions and practices.**

Social ethics of rural farming communities' contribution to EBP in agriculture extension information services: Through interviews, the findings indicate that farmers had a negative attitude and poor mindset about any new evidence in any form, including technology, which they perceive as foreign and unfamiliar. They preferred friends who were familiar with and spoke the same local language as sources of information. The rural farmers had a challenge with language and preferred information aggregated. The same findings revealed that translation and repackaging of the EBI were not performed at the district level but at the Ministry level, which gave the rural people little input into this process for ownership. These findings suggest that the social value of the rural community should be considered in the agriculture extension information services process for an effective EBP.

## Discussion

### Sources of agriculture EBI

This study examined the various agricultural sources of information on maize postharvest handling available to rural agriculture extension officers and farmers. Participants in the personal interviews, focus groups and the respondents from the survey agreed that agriculture extension officers had multiple sources of evidence, which included the MAAIF, the Internet,

AGORA, and partners such as the National Agriculture Research Organisation (NARO) and the World Food Programme (WFP). These findings mean that EBP for agriculture extension information services is possible. The mention of research sources like NARO and AGORA meant that scientifically researched evidence was accessed and channelled to the farmers.

Agriculture extension officers used training, farm visits and demonstrations, radio talk shows, WhatsApp, farm explorers and exhibitions to disseminate this information to rural smallholder farmers. These findings imply that these officers had established channels they could use to ask for specific information needs from farmers and assess the application of the evidence disseminated through the farm visits and demonstrations. Furthermore, WhatsApp implies that mobile technology, including smartphones and Internet connectivity, was also available in the rural areas studied. However, the Agriculture Office, the Internet, WhatsApp, and community libraries were the least used sources compared to friends. These findings could mean that these facilities were accessible to a few people. Moreover, given that most rural smallholder farmers accessed information from unofficial sources (i.e., friends), it was difficult to verify the quality of the information and assess its effectiveness in influencing the right mindset, decisions and practices in maize postharvest handling.

People generally prefer sources of information they know and feel they can trust to provide information that is easy to access/use and available. Friends being the most preferred channel of information for rural farmers was consistent with previous studies that showed rural smallholder farmers preferred informal over formal information sources [76–78]. A study from Spain reported that 85% of farmers relied on personal experience as their information source, 70% had low levels of education and agriculture training, and information service seeking was considered a private rather than a public affair [79]. Therefore, accessing and using information depends on the users' level of education, information literacy levels [80] and exposure to sources of information.

Consequently, this study's findings that five kinds of evidence (tacit, explicit, farmers' experiences, equipment, and technological materials) [81] were disseminated to rural farmers agree with those reported in other studies [3,82,83];). The findings suggested that the sources of maize postharvest handling information needed update and made adequately enough for agriculture extension officers and farmers. Access to reliable and up-to-date information is crucial for improved agricultural extension services. Gohain [84] highlighted that information was an important component of agricultural growth, and blending indigenous and scientific knowledge sources was crucial for agriculture sustainability.

## Knowledge and skills in information and communication technologies (ICT)

The findings from both the qualitative and quantitative datasets suggested that ICTs had not reached all parts of Uganda, and some rural smallholder farmers could not access EBI online, even via WhatsApp. The document analysis showed that the Uganda National Extension Strategy 2016–20/21 [85] recognised the capabilities of digital information and ICTs to improve extension services. However, this strategy was not fully implemented for agriculture extension information services, specifically for maize farmers. These findings were consistent with previous studies that reported ICTs were poor, costly or non-existent in some rural areas in most developing countries [5,59,86]

These findings indicated that ICTs should be improved for rural agriculture extension services to enable rural agriculture extension officers and farmers in rural areas to access EBI and practice EBP more effectively. Extension workers should use ICTs, including AI-enabled technologies like chatbots, blockchain, and AI-generated content (AIGC), to package and disseminate EBI as suggested by Hussain, Batool [87]; Liu, Kong [48]; Maginga, Kutuku [29]; Maria [28]; Raj, Masih [27]; TuYizere, Uwase [26]; Zhang, Zhao [52] while considering the rural community social values and ethics [49]. Mobile technology for agricultural advice has shifted farmers away from less reliable sources, fostering better management practices [88–90]. The ability to effectively use ICT tools such as databases, mobile apps, and online platforms influences the implementation of EBP. Extension officers with strong ICT skills can access, analyze, and share information more quickly and accurately. Additionally, ICT can facilitate communication with farmers, enabling the timely dissemination of evidence-based advice. Lack of ICT proficiency may result in missed opportunities to leverage digital tools for improving practice and extending new knowledge to the farming community.

**Evidence-based information practices and agriculture practice of maize postharvest handling**

This study's findings suggest that the rural smallholder farmers, the target for agricultural extension services, need their maize postharvest handling problems studied (asked) to determine what EBI they need to acquire. The information acquired should be appraised for relevance and quality to suit rural smallholder farmers' needs and solve their problems. The information should also be aggregated (i.e., translated and packaged) for easy access and use by rural farmers in their social context. Finally, the applied EBIP must be monitored and assessed for feedback before the subsequent dissemination to correct errors and measure effectiveness. All these expressions from the study participants show that the EBP was necessary. EBP theory states that evidence should be checked for quality [91] and relevance to produce the desired goals through the implementation of EBP. This idea was proposed by EBP scholars as the '6As' (ask, acquire, appraise, aggregate, apply and assess) [36,91–93].

The significant relationship between the application of evidence and the maize postharvest handling decisions and practices reported in this study suggested that EBIP had a role in accessing and applying quality, credible, and relevant information. If well implemented, quality and relevant information can mitigate the problem of the information that smallholder farmers receive not being valid for post-harvest handling [4]. Understanding the importance of EBP encourages extension officers to verify the information disseminated by farmers. This understanding starts with awareness fostered through workshop training and exposure to successful case studies where evidence-based methods have improved extension services. Agricultural extension officers who implement structured, evidence-based information practices ensure that their maize postharvest advice and interventions to farmers are grounded in scientific and verifiable information. When such kind of information is accessed and used, it will influence the mindset for effective practice, and eventually, postharvest loss will be reduced. Engaging local farmers when designing and implementing extension information service programs may make the services contextually relevant and more likely to succeed [27,90].

The qualitative participants reported that the EBIP was inadequate in training, knowledge, and skills, the top-to-bottom approach to providing evidence was not suitable, the lack of evidence, and the personnel mandated to disseminate EBI. The personal interviews suggested that the MAAIF and other stakeholders, such as farmer groups, NARO and the WFP, should work together to harmonise, retool and update extension officers with ICT knowledge and skills for online access, use, packaging and dissemination of relevant maize postharvest EBI. This would improve farmers' mindsets and practices to alleviate loss and improve food security and household incomes for rural smallholder farmers. A similar suggestion was highlighted in previous studies [78,94,95]. The goal of accessing and disseminating evidence-based information on maize postharvest handling practices is to ensure unfavourable practices are minimised for the sustainability of agriculture, food security, and improved human life in farming communities.

## Conclusion

This study found that agriculture extension officers leveraged digital evidence-based information practices for sustainable agricultural advisory services to rural farmers, though not officially documented. Challenges were identified, like inadequate knowledge and skills, insufficient access to updated evidence, and inadequate resources, including budget support to implement evidence-based information practices. The rural agriculture information officers needed updated knowledge and skills in evidence-based practice, a source of evidence, information literacy and ICT training to sustain support offered to farmers to reduce postharvest loss and enhance food security in four dimensions: availability, access, utilisation and sustainability.

## Recommendations and study implications

To address these shortcomings, the study recommends implementing continuous professional development (CPD) programs for agricultural extension officers, with a focus on information literacy, digital evidence-based practice, ICTs

for digital evidence, translation and repackaging of evidence for rural communities that recognise their social and ethical values for sustaining effective services to rural communities. These programs would help agriculture extension officers nurture the necessary desire and skills to use digital tools and platforms to access and disseminate relevant agricultural information for rural farming communities to enhance postharvest handling practices for food security and income. Information literacy training should be a core component of these CPD programs, delivered through seminars, workshops, and short-term training sessions. Such training would equip extension workers to critically evaluate, select, and share high-quality, maize postharvest handling evidence-based information with farmers, ensuring that the information is credible and relevant.

Incorporating information literacy into CPD initiatives would also enable extension officers to guide farmers in using digital resources effectively, ensuring that farmers receive information and are empowered to apply it to improve their post-harvest practices, food security and household income. This approach would enhance the overall effectiveness of EBIP by improving the knowledge dissemination process and ensuring the information is accessible, understandable, and actionable for farmers. By integrating these training components into CPD programs, extension officers will be better positioned to implement the 6As framework (ask, acquire, appraise, aggregate, apply, and assess) and provide continuous, high-quality support to farmers.

## Supporting information

**S1 Appendix. Study questionnaire.**
(DOCX)

## Author contributions

**Conceptualization:** Jackline Estomihi Mayende Kiwelu, Patrick Ngulube.

**Data curation:** Jackline Estomihi Mayende Kiwelu.

**Formal analysis:** Jackline Estomihi Mayende Kiwelu.

**Investigation:** Jackline Estomihi Mayende Kiwelu.

**Methodology:** Jackline Estomihi Mayende Kiwelu.

**Project administration:** Jackline Estomihi Mayende Kiwelu.

**Resources:** Jackline Estomihi Mayende Kiwelu.

**Software:** Jackline Estomihi Mayende Kiwelu.

**Supervision:** Patrick Ngulube.

**Validation:** Jackline Estomihi Mayende Kiwelu.

**Visualization:** Jackline Estomihi Mayende Kiwelu.

**Writing – original draft:** Jackline Estomihi Mayende Kiwelu.

**Writing – review & editing:** Jackline Estomihi Mayende Kiwelu, Patrick Ngulube.

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
