## [Decision Letter · Decision Letter 0]

17 Sep 2025

Dear Dr. Kiwelu,

Thank you for submitting your manuscript to PLOS ONE. After careful consideration, we feel that it has merit but does not fully meet PLOS ONE’s publication criteria as it currently stands. Therefore, we invite you to submit a revised version of the manuscript that addresses the points raised during the review process.

We look forward to receiving your revised manuscript.

Kind regards,

Charles Odilichukwu R. Okpala, PhD

Academic Editor

PLOS ONE

Journal Requirements:

4. Please include a copy of Table 6 which you refer to in your text on page 29.

Additional Editor Comments :

Please, kindly address all comments raised in detail

Reviewers' comments:

Reviewer's Responses to Questions

**Comments to the Author**

1. Is the manuscript technically sound, and do the data support the conclusions?

Reviewer #1: Partly

Reviewer #2: Yes

2. Has the statistical analysis been performed appropriately and rigorously?

Reviewer #1: No

Reviewer #2: Yes

3. Have the authors made all data underlying the findings in their manuscript fully available?

Reviewer #1: No

Reviewer #2: Yes

4. Is the manuscript presented in an intelligible fashion and written in standard English?

Reviewer #1: Yes

Reviewer #2: Yes

Reviewer #1: Abstract Findings: The authors indicate the most used evidence-based practices were “asking, acquiring, appraising, and applying,” and that “aggression and assessment” were less practiced. These are words without any context and require the reader to know what the authors are talking about. There needs to be more stated about what the authors mean by these words.

Page 6, Discussion about Postharvest Handling: The authors indicate there are problems with postharvest handling in Uganda, such as mold, termites, and aflatoxin contamination that undermines maize quality. However, the authors say nothing about how maize is handled postharvest in Uganda. What are the typical harvest practices used by maize farmers in Uganda? How do farmers transport their maize crops to market? What types of storage facilities are available for storing maize? What types of markets accept the maize crop, and where are they located. Do maize farmers have to transport their crops for long distances? The problem statement is too general. More specifics need to be added to properly address the postharvest handling problems the authors refer to in their paper.

Page 12, Artificial Intelligence Discussion: The first sentence discussing AI needs to be rewritten for better clarity.

Figure 1: This figure is very hard to read in its present form. The words in the boxes are blurry.

Page 16, Hypothesis Test Section: This section is repetitive (same words for each EBIP) and can be summarized by stating the authors were testing that no significant relationship existed for each of the 6As. Also, this would be the place for the authors to state the type of method used to for hypothesis testing in their study.

Page 17, Reference to Using a Questionnaire: The authors indicated they used a questionnaire to gather quantitative data. The authors give no specifics about the types of questions asked in the questionnaire, or if the questionnaire was administered to both agricultural extension officers and rural maize smallholder farmers or just to maize farmers. It would be good for the authors to supply the questionnaire instrument in an appendix to the manuscript or at least state what types of questions were asked somewhere in the manuscript.

Page 19, ATLAS.ti 24: The authors indicated they used ATLAS.ti 24. What is ATLAS.ti 24?

Page 21, FGD: This acronym needs to be spelled out, as the reader does not know what “FSD” is.

Page 22, Table 2 and Page 26, Table 3: What is “SACCO”? This acronym needs to be spelled out in a footnote to Tables 2 and 3.

Page 10, the “6As” and Page 27, Table 4: As with the abstract, these evidence-based information practices are very general with no context. In Table 4, how were these EBIPs quantified? What do the numbers in the “Number” column of Table 4 mean? Do the numbers represent maize farmers who responded that rural extension officers actually performed each listed EBIP? If so, what types of questions were asked of maize farmers to acquire this information for each EBIP?

Figure 4: This figure is very blurry.

Table 5, Page 28: Again, how were the numbers in the “Number” column quantified? Also, the numbers reported in Table 5 are virtually the same as those reported in Table 4. Is there a need for two tables? Why not combine these tables?

Reviewer #2: I appreciate the efforts of the researchers, the paper contains adequate points to address the problem. however, the introduction is extended and must be minimized. the paper is well written and touches the important concern of the research under investigation.

**Do you want your identity to be public for this peer review?** For information about this choice, including consent withdrawal, please see our Privacy Policy

Reviewer #1: No

Reviewer #2: **Yes: ** Shishay Kahsay Weldearegay

---

## [Author Response · Author response to Decision Letter 1]

6 Oct 2025

The response to the editors and the reviewers are attached as directed by the editor under the name " Response to reviewers".

---

## [Decision Letter · Decision Letter 1]

22 Oct 2025

Enhancing food security sustainability through digital information extension services in rural Uganda: maize postharvest evidence-based strategies

PONE-D-25-39333R1

Dear Dr. Kiwelu,

We’re pleased to inform you that your manuscript has been judged scientifically suitable for publication and will be formally accepted for publication once it meets all outstanding technical requirements.

Kind regards,

Charles Odilichukwu R. Okpala, PhD

Academic Editor

PLOS ONE

Additional Editor Comments (optional):

Acceptable for publication

Reviewers' comments:

Reviewer's Responses to Questions

**Comments to the Author**

Reviewer #1: All comments have been addressed

Reviewer #2: All comments have been addressed

2. Is the manuscript technically sound, and do the data support the conclusions?

Reviewer #1: Yes

Reviewer #2: Yes

3. Has the statistical analysis been performed appropriately and rigorously?

Reviewer #1: (No Response)

Reviewer #2: Yes

4. Have the authors made all data underlying the findings in their manuscript fully available?

Reviewer #1: Yes

Reviewer #2: Yes

5. Is the manuscript presented in an intelligible fashion and written in standard English?

Reviewer #1: Yes

Reviewer #2: Yes

Reviewer #1: I am satisfied with all the revisions made by the Authors. I have no further reservations or questions

Reviewer #2: I went through the paper, the author made significant improvement to the research article. my concerns are well addressed and aligns with the PLOS One data policy.

**Do you want your identity to be public for this peer review?** For information about this choice, including consent withdrawal, please see our Privacy Policy

Reviewer #1: No

Reviewer #2: **Yes: ** Shishay Kahsay Weldearegay

---

## [Editor Report · Acceptance letter]

PONE-D-25-39333R1

PLOS ONE

Dear Dr. Kiwelu,

I'm pleased to inform you that your manuscript has been deemed suitable for publication in PLOS ONE. Congratulations! Your manuscript is now being handed over to our production team.

Kind regards,

on behalf of

Dr. Charles Odilichukwu R. Okpala

Academic Editor

PLOS ONE